# Implementation of Computed Tomography Angiography (CTA) and Computed Tomography Perfusion (CTP) in Polish Guidelines for Determination of Cerebral Circulatory Arrest (CCA) during Brain Death/Death by Neurological Criteria (BD/DNC) Diagnosis Procedure

**DOI:** 10.3390/jcm10184237

**Published:** 2021-09-18

**Authors:** Romuald Bohatyrewicz, Joanna Pastuszka, Wojciech Walas, Katarzyna Chamier-Cieminska, Wojciech Poncyljusz, Wojciech Dabrowski, Joanna Wojczal, Piotr Luchowski, Maciej Guzinski, Elzbieta Jurkiewicz, Monika Bekiesinska-Figatowska, Radoslaw Owczuk, Jerzy Walecki, Olgierd Rowinski, Maciej Zukowski, Krzysztof Kusza, Mariusz Piechota, Andrzej Piotrowski, Marek Migdal, Marzena Zielinska, Marcin Sawicki

**Affiliations:** 1Department of Anesthesiology and Intensive Therapy, Pomeranian Medical University, 70-204 Szczecin, Poland; romuald.bohatyrewicz@pum.edu.pl; 2Pediatric and Neonatal Intensive Care Unit, Institute of Medical Science, University of Opole, 45-040 Opole, Poland; wojciechwalas@wp.pl; 3Department of Diagnostic Imaging and Interventional Radiology, Pomeranian Medical University, 70-204 Szczecin, Poland; kac.iiafix@gmail.com (K.C.-C.); wojciech.poncyljusz@pum.edu.pl (W.P.); msaw3108@gmail.com (M.S.); 4Department of Anesthesiology and Intensive Care, University of Lublin, 20-059 Lublin, Poland; w.dabrowski5@gmail.com; 5Department of Neurology, Medical University of Lublin, 20-954 Lublin, Poland; jwojczal@poczta.onet.pl (J.W.); pluchowski@wp.pl (P.L.); 6Department of General Radiology, Interventional Radiology and Neuroradiology, Wroclaw Medical University, 50-367 Wroclaw, Poland; guziol@wp.pl; 7Department of Diagnostic Imaging, The Children’s Memorial Health Institute, 04-730 Warsaw, Poland; E.Jurkiewicz@IPCZD.PL (E.J.); a.piotrowski@ipczd.pl (A.P.); m.migdal@IPCZD.PL (M.M.); 8Department of Diagnostic Imaging, Institute of Mother and Child, 01-211 Warsaw, Poland; monika.bekiesinska@imid.med.pl; 9Department of Anesthesiology and Intensive Therapy, Faculty of Medicine, Medical University of Gdansk, 80-210 Gdansk, Poland; r.owczuk@gumed.edu.pl; 10Diagnostic Radiology Department, Central Clinical Hospital of the Ministry of the Interior in Warsaw, 02-507 Warsaw, Poland; jerzywalecki@o2.pl; 11Department of Clinical Radiology, Medical University of Warsaw, 02-091 Warsaw, Poland; olgierd.rowinski@wum.edu.pl; 12Departament of Anesthesiology Intensive Care and Acute Poisoning, Pomeranian Medical University, 70-204 Szczecin, Poland; zukowski@pum.edu.pl; 13Department of Anaesthesiology and Intensive Therapy, Poznan University of Medical Sciences, 61-701 Poznan, Poland; k-kusza@wp.pl; 14Department of Anaesthesiology and Intensive Therapy, Centre for Artificial Extracorporeal Kidney and Liver Support, Dr W. Bieganski Regional Specialist Hospital, 91-347 Lodz, Poland; mariuszpiechota@poczta.onet.pl; 15Department of Anesthesiology and Intensive Therapy, Wroclaw Medical University, 50-367 Wroclaw, Poland; marzena.zielinska@umed.wroc.pl

**Keywords:** brain death, death by neurologic criteria, cerebral blood flow, CT angiography, CT perfusion

## Abstract

Background: Brain death/death by neurologic criteria (BD/DNC) guidelines are routinely analyzed, compared and updated in the majority of countries and are later implemented as national criteria. At the same time, extensive works have been conducted in order to unify clinical procedures and to validate and implement new technologies into a panel of ancillary tests. Recently evaluated computed tomography angiography and computed tomography perfusion (CTA/CTP) seem to be superior to traditionally used digital subtraction angiography (DSA), transcranial Doppler (TCD) and cerebral perfusion scintigraphy for diagnosis of cerebral circulatory arrest (CCA). In this narrative review, we would like to demonstrate scientific evidence supporting the implementation of CTA/CTP in Polish guidelines for BD/DNC diagnosis. Research and implementation process: In the first of our base studies concerning the potential usefulness of CTA/CTP for the confirmation of CCA during BD/DNC diagnosis procedures, we showed a sensitivity of 96.3% of CTA in a group of 82 patients. CTA was validated against DSA in this report. In the second study, CTA showed a sensitivity of 86% and CTP showed a sensitivity of 100% in a group of 50 patients. In this study, CTA and CTP were validated against clinical diagnosis of BD/DNC supported by TCD. Additionally, we propose our CCA criteria for CTP test, which are based on ascertainment of cerebral blood flow (CBF) < 10 mL/100 g/min and cerebral blood volume < 1 mL/100 g in regions of interest (ROIs) localized in all brain regions. Based on our research results, CTA/CTP methods were implemented in Polish BD/DNC criteria. To our knowledge, CTP was implemented for the first time in national guidelines. Conclusions: CTA and CTP-derived CTA might be in future the tests of choice for CCA diagnosis, proper and/or Doppler pretest might significantly increase sensitivity of CTA in CCA diagnosis procedures. Whole brain CTP might be decisive in some cases of inconclusive CTA. Implementation of CTA/CTP in the Polish BD/DNC diagnosis guidelines does not show any major obstacles. We believe that in next edition of “The World Brain Death Project” CTA and CTP will be recommended as ancillary tests of choice for CCA confirmation during BD/DNC diagnosis procedures.

## 1. Background

In this narrative review, we present the evolution of Polish guidelines for brain death/death by neurologic criteria (BD/DNC) diagnosis and demonstrate scientific evidence supporting the implementation of computed tomography angiography (CTA) and computed tomography perfusion (CTP) for confirmation of cerebral circulatory arrest (CCA).

The first Polish BD/DNC criteria implemented in 1984 as whole-brain death criteria were similar to the so-called “Harvard brain death criteria” [1]. Later, these Polish criteria underwent a unique evolution: in 1990, they were converted into brainstem death criteria and, later, in 2007, they were reversed back to whole-brain death criteria [2], subsequently being amended in 2020 [3]. During this conversion, we reimplemented an opportunity for facultative usage of instrumental ancillary tests, including brain blood perfusion tests.

Currently, in Poland, although formally facultative, instrumental ancillary tests are used in the majority of BD/DNC diagnosis procedures. This is exactly the reverse of what is proposed in current international recommendations [4]. There are various reasons for this. Despite receiving the official support of health care authorities and three consecutive Popes, we have faced unclear or even negative statements from some religious commentors and selected Catholic media. The matter is sometimes complicated by political activities, which induce doctors’ uncertainty and filling of a lack of legal safety. Polish physicians would feel more comfortable and safer if they could demonstrate the results of instrumental ancillary tests confirming BD diagnosis. Brain blood perfusion tests are considered to be the most evident from all sides of the discussion—physicians, relatives of the deceased and public opinion. It is obvious to everybody, regardless of the level of education, that non-perfused tissue must be dead.

There are three brain blood flow ancillary tests that, despite known disadvantages, have a long-established position in BD/DNC diagnosis:Catheter digital subtraction angiography (DSA), selective or from the aortic arch;Transcranial Doppler ultrasonography (TCD);Cerebral perfusion scintigraphy.

Use of DSA, which is still considered to be a gold standard and reference method, is gradually decreasing [4]. It is invasive and requires procedural skills, time and availability of an angiography suite. It was reimplemented in Polish BD criteria in 2007 and was approved again in the amendment in 2020 [3]. TCD is completely noninvasive, its sensitivity exceeds 90% and the specificity is 100%. Despite its obvious advantages, TCD is not frequently used in Poland because it has to be performed with unique devices that, currently, are not widely available. Additionally, it is highly operator-dependent and requires certification of the performing physician. The last of the “traditional” cerebral perfusion tests, scintigraphy, is currently seldom available in the majority of Polish hospitals.

In this difficult situation, it has become of crucial importance to develop and introduce an ancillary test that would be less invasive than DSA, easily available, uncomplicated to perform and easy to interpret.

## 2. Early Research into Using CTA/CTP for Diagnosing BD/DNC in Poland

In 2005, we considered CTA and CTP to be the tests of choice for the future, which is consistent with the current opinion of Greer et al. [4]. Modern multi-slice CT scanners are fast enough to visualize vasculature and perfusion of the whole brain with a single intravenous injection of iodinated contrast medium and, finally, to confirm CCA. Both CTA and CTP, if performed with a calibrated intravenous contrast injection and precise scanning protocols, are operator-independent at the stage of performance and providing raw data.

The research team of the Department of Anesthesiology and Intensive Care together with co-workers from the Department of Diagnostic Imaging and Interventional Radiology of the Pomeranian Medical University in Szczecin have been involved in research programs and legislation since the implementation of the first Polish BD/DNC criteria published in 1984. In 2005–2007, Romuald Bohatyrewicz was a co-chairman and in 2015–2019, chairman, of the Ministry of Health’s Task Force for review of these criteria.

In 2007, we suggested the possibility of implementing CTA as a new brain blood flow test, but this was not accepted by the rest of the Task Force members because of insufficient evidence in the literature and lack of experience in Poland. In this situation, we organized a national multi-center trial (N N403 171137), entitled “Evaluation of CT angiography and CT perfusion in brain death diagnosis”, in a group of adult brain-dead patients, followed by a series of publications. The first of them, published in 2010 [5], confirmed the ability of CTA/CTP to diagnose CCA in our population of BD/DNC patients and our findings were compatible with data published by other authors at that time [6,7,8,9,10]. Unfortunately, our CTP findings could not be applied to CCA diagnosis because the generation of CT scanners used at that time in Poland could cover only a thin layer of the brain, about 30 mm thick.

During initial attempts to implement CTA for CCA diagnosis, there was no consensus regarding the evaluation criteria, which have evolved as research progressed [6,7,8,9,10,11,12,13]. The most popular scoring systems (10, 7, 4 points), shown in Figure 1, were based on analysis of the opacification of the following:Pericallosal segments of the right and left anterior cerebral artery (ACA-A3);Cortical segments of the right and left middle cerebral artery (MCA-M4);Cortical segments of the right and left posterior cerebral artery (PCA-P2);Basilar artery (BA);Right and left internal cerebral vein (ICV);Great cerebral vein (GCV)—the vein of Galen.

Initially, we analyzed CTA imaging in a group of 82 patients undergoing routine BD/DNC diagnosis with DSA included as a standard element of this procedure. CTA was completed first and was followed by DSA [14]. In this situation, CTA could be validated against DSA with a very short time interval between these two procedures, which is consistent with the recommendation for method validation recently published by Greer et al. [4]. The sensitivity reported in this study reached 96.3% according to the 4-point scale, 74.4% according to the 7-point scale and 67.1% according to the 10-point scale [14].

After meticulous analysis of these data, we finally accepted the 4-point scale proposed by the French guidelines for diagnosis of BD/DNC [15]. According to this 4-point scale, CCA may be confirmed if there is a bilateral absence of contrast filling of cortical segments of the middle cerebral arteries (MCA-M4) and internal cerebral veins (ICVs), as presented in Figure 1. Unilateral opacification of one or two cortical branches of the MCA does not preclude the diagnosis of CCA as long as the contrast does not fill the ICVs.

Additionally, we noticed (unpublished results) that if CTA tests were performed shortly after the appearance of brainstem areflexia, more widespread opacification of cerebral vessels, thus excluding CCA diagnosis, was recorded, which was also confirmed by the data published by Welschehold and Kerhuel [13,16]. The explanation for this phenomenon is quite simple: in this short period, the intracranial pressure (ICP) did not exceed the mean arterial pressure (MAP), leading to CCA. Premature CTA examinations lead to a dramatic decrease in test sensitivity, which is one of the sources of undeserved opinions about the poor utility of this method for CCA determination. Such cases are rather unlikely in Poland because our doctors prefer more conservative approaches and generally initiate diagnostic procedures after a longer observation period. Nevertheless, in the Polish guidelines [3], we recommended a minimal 6-h observation time before CTA/CTP examinations, which is identical to the French guidelines [15].

## 3. Comparison with the Other Instructions for CCA Confirmation by CTA Imaging

Recently Lewis et al. reviewed diagnostic requirements for ancillary testing for BD/DNC in 78 official national BD/DNC protocols and found that in 14 European countries CTA was included in to a panel of ancillary tests [17], but according to our knowledge this method was in detail described in national guidelines and relatively frequently used only in France, Germany and recently in Poland. French and Polish protocols and diagnostic criteria based on 4-point scale are similar while German protocol elaborated on the grounds of publication of Welschehold [13] is based on recognition of lack of opacification of 7 intracranial arteries in late arterial phase [18]. Comparison of these three diagnostic protocols is demonstrated in Table 1.

## 4. Sensitivity and Specificity of CTA in CCA Determination during BD/DNC Diagnosis Procedures

Test accuracy is defined by two important factors, sensitivity and specificity. The sensitivity of CTA and CTP in BD/DNC diagnostic procedures refers to their ability to correctly indicate BD/DNC in patients with true BD/DNC. In our studies, it would be the proportion of positive (confirmed) CCA in a group with confirmed BD/DNC diagnosis. Specificity would relate to the test’s ability to correctly identify patients without the BD/DNC. In our case this would be the proportion of patients without recognized CCA (negative) in a group of patients who are not brain dead. In our studies evaluating CTA the reference was DSA while in those evaluating CTP we used clinical diagnosis as the reference. A test with a sensitivity of around 90% would be considered to have good diagnostic performance. Obviously, in this special clinical situation, the aim should be to achieve 100% specificity.

Assessment of accuracy of the test requires the established ground truth as a reference. However, in previous studies evaluating CTA and CTP in the BD/DNC diagnostic procedure different reference standards were used, e.g., clinical signs of BD/DNC (most commonly), DSA, perfusion scintigraphy or TCD. This is one of causes of significant divergence in reported sensitivities. Therefore, we agree with the authors of „World Brain Death Project” [4] that establishing unified reference for studies evaluating CTA and CTP in the BD/DNC diagnostic procedure is desirable.

According to information available in “World Brain Death Project” concerning blood perfusion tests, their sensitivity varies mainly in a range of 52–100% while declared specificity is close to 100% with remark included: “Specificity is assumed on basis of experimental data but should be interpreted with caution given the limitation of studies that reported only on clinically confirmed BD/DNC” [4]. We support this opinion.

We did not determine specificity of CTA/CTP examination during BD/DNC diagnostic procedures which in fact is one of the limitations of our studies. In the context of our research it would concern the frequency of false positive diagnoses of CCA, potentially supporting incorrect BD/DNC diagnosis. Such situation would be catastrophic in case of BD/DNC diagnosis in a patient with survival and recovery potential which is also highlighted in “World Brain Death Project”. Fortunately, such theoretical situation is unlikely because all ancillary tests are only additional tools used in complex procedure of BD/DNC diagnosis process including assessment of devastating brain injury, analysis of preconditions and prerequisites and, finally, meticulous clinical examination [4]. In case of any doubts termination of BD/DNC diagnosis process is worldwide recommended. This happened in a few reported cases of patients demonstrating brainstem areflexia with persisted respiratory drive and CCA diagnosed by DSA [19,20]. However, no one of these patients survived, but such discrepancy between instrumental test and clinical findings might be highly confusing. This might be explained by the fact, that infratentorial space, especially medulla oblongata is surrounded by osseous and obscured by highly vascularized structures. In some cases, it may be supplied by the posterior inferior cerebellar artery atypically originating from extracranial segment of vertebral artery [19]. Vestigial blood supply sufficient to preserve at least minimal partial function of respiratory centre might remain undetected by any of blood flow tests. Considering this, specificity of 100% is not achievable in any one of the blood flow studies.

Nevertheless, specificity remains problematic during assessment of brain blood perfusion tests because patients non suspected but close to develop BD/DNC are rarely included in such studies. However, it would be possible to identify them in a group of patients hospitalized in centers where neurointerventional procedures are carried out on regular basis. We included such 5 participants in one of our publications [21]. We found only one publication by Welschehold et al. [13] elaborating prospectively the issue of CTA specificity during CCA diagnostic procedures. He performed CTA in 30 patients immediately after the first signs of loss of brainstem reflexes were noticed and a few hours later, after definitive legal determination of BD/DNC [13]. He found CCA in 3 out of 30 patients short after onset of brainstem areflexia but before legal determination of BD/DNC and interpreted them as false positives. All of these 3 patients were later legally declared brain dead. CCA appeared in this group in fact earlier than in remaining 27 cases and considering findings of these 3 patients as false positives, although formally justified, is slightly unfortunate.

Majority of questions concerning sensitivity and specificity of CTA/CTP tests as well as CCA dynamics will be answered by extensive prospective multicenter trial NCT0309851 initiated in 2017 by Chassé and Shankar in Canada. They planned enrollment of 333 participants, with high risk of BD/DNC, not in a course of BD/DNC diagnosis at that moment. The study is oriented for determination of CTA/CTP accuracy in BD/DNC procedures with special attention for diagnosis of brainstem hypoperfusion. We were discussing with Canadian Colleagues possible Polish multicenter participation in this study, but after meticulous analysis of the project we realized, that Polish Bioethical Committee would not accept invasive tests in patients who do not demonstrate complete brainstem areflexia as being not in the best interest of patients and in the same time useless for potential BD/DNC diagnosis. Therefore, finally, we did not join the trial.

## 5. Research Advancement with CTP in Poland

At the same time, we noticed that in rare cases of patients demonstrating BD/DNC symptoms, preserved trace opacification of intracranial arteries may be observed in DSA examination. This phenomenon is known as stasis filling, defined as delayed, weak and persistent opacification of the proximal cerebral arterial segments, without opacification of the cortical branches or venous outflow [22]. In these rare cases, CTP often shows residual cerebral blood flow (CBF) below 10 mL/100 g/min and a cerebral blood volume (CBV) below 1.0 mL/100 g. These values are the established thresholds for neuronal necrosis [23] and following this, they may be considered thresholds for global or regional CCA diagnosis [24,25].

Advanced research activity concerning CTP became possible after the advent of a new generation of CT scanners fast enough to visualize vasculature and perfusion of the whole brain with a single intravenous injection of iodinated contrast medium. At that time, we stopped performing CTA imaging and switched to reconstruction of CTA images from the CTP source images as timing-invariant (TI)-CTA. TI-CTA provides angiography by overlapping all time frames and displaying the maximum enhancement over time. This makes the technique time independent, which means that the maximum enhancement of a vessel is displayed independently of contrast arrival time. Therefore, TI-CTA is not sensitive to delayed arrival of the contrast material in cerebral vessels and, thus, should display any vessel present. This technique was previously described and shown to be reliable by Smit et al. [26].

CTP criteria for CCA during the BD/DNC diagnostic procedure were not published before; therefore, we elaborated our original instruction of assessment based on an analysis of the CBF and CBV in 1-cm^2^ circular regions of interest (ROIs), including the midbrain (two ROIs), the pons (two ROIs) and the medulla oblongata (two ROIs) as well as the cerebellum (eight ROIs); cortical regions of the frontal (12 ROIs), parietal (12 ROIs), temporal (12 ROIs) and occipital lobes (eight ROIs); and the basal ganglia (eight ROIs), drawn bilaterally and placed on each 10-mm axial slice, as shown in Figure 2. We recognized CCA in CTP examination if the CBF value was below 10 mL/100 g/min and CBV was below 1.0 mL/100 g in all ROIs [25]. The most frequent combinations of CTA and CTP images are shown in Figure 3.

In the next step of our research program, we hypothesized that CTP would be a more sensitive approach than CTA in CCA diagnosis. To verify this, we conducted a study aiming to compare the sensitivities of CTP and CTA in recognizing CCA during BD/DNC diagnosis procedures. A group of 50 patients undergoing this diagnostic procedure were included in the study. All of them met the standard BD/DNC criteria based on confirmation of catastrophic brain injury, exclusion of confounders and confirmation of brainstem areflexia and apnea during two series of clinical examinations [25]. Additionally, TCD examination confirming CCA was completed in the majority of them; however, this information was not included in the publication.

In 43 out of 50 patients, CTA confirmed CCA, as demonstrated in Figure 3, patient B. In the remaining seven patients, CTA revealed opacification of M4 segments, ICVs or both, as shown in Figure 3, patients C and D. These CTA findings were inconsistent with CCA according to the 4-point scale. In all 50 patients, CBF was below 10 mL/100 g/min and CBV was below 1.0 mL/100 g, which confirmed CCA according to our criteria. In summary, in this publication, we reported a sensitivity of 86% and 100% for CTA and CTP, respectively. Additionally, these results confirmed our hypothesis that in borderline cases, when CTA is inconclusive, CTP may be a decisive method for CCA diagnosis. Later, after analysis of a few additional cases, we stated that in special situations in patients with clinical signs of BD, isolated areas of decompression may be preserved in the region of craniectomy or open fractures. These isolated areas may exhibit CBF and CBV values above the thresholds of 10 mL/100 g/min and 1 mL/100 g, respectively. This phenomenon does not exclude the diagnosis of CCA if the CBF and CBV values are below these thresholds in other ROIs, including those in the brainstem, as shown in Figure 3, patient E. Appearance of multiple areas with CBF and CBV above the threshold values is, according to our current opinion, inconsistent with CCA diagnosis, as shown in Figure 3, patient F.

Noteworthily, another research group led by Shankar [24,27] postulated a slightly different approach to the possible usefulness of CTP imaging in BD/DNC diagnostic procedures or withdrawal of life-sustaining therapy. They focused on demonstration of brain hypoperfusion limited to the brainstem area, which is, in fact, somehow parallel to brainstem areflexia and the brainstem death concept, suggesting “isolated brainstem death”. CTP in this situation does not necessarily confirm global CCA as it was demonstrated in our report [25]. It is highly questionable whether isolated brainstem death diagnosis confirmed by CTP imaging, but coexisting with preserved supratentorial perfusion and possibly persisted EEG activity, might justify BD/DNC diagnosis.

## 6. Implementation of CTA/CTP Examination into the Polish National Guidelines for BD/DNC

Based on our published research data [14,25] and unpublished observations, finally, we implemented CTA/CTP examination in the Polish national guidelines for BD/DNC diagnosis in patients over 12 years of age at the beginning of 2020 [3]. To our knowledge, this was the first implementation of CTP in official BD/DNC diagnosis guidelines. A diagram showing two alternative CT diagnostic approaches depending on the technical capabilities of the scanner, local tradition and the radiologist’s competence is demonstrated in Figure 4.

CTA/CTP examinations for BD/DNC diagnosis procedures were approved for patients > 12 years old, assuming that above this age, the skull is not pliable and the brain reaches morphological maturity. We presumed that the mechanisms of CCA might be similar in younger age groups, at least >2 years old without patent sutures or fontanels and in patients < 2 years old, according to our knowledge, it is unpredictable, as with the results of other traditional methods used for CCA diagnosis [4]. Therefore, in order to explore this issue, we recently invited all Polish and foreign pediatric intensive care units to participate in a multi-center study for the validation of CTA and CTP in determination of CCA during the BD/DNC diagnosis procedure in a pediatric population below 12 years of age [28].

Immediately after the introduction of CTA/CTP for CCA diagnosis into Polish guidelines, we started monitoring the usage of these methods all over the country. Unfortunately, this was extremely difficult due to the COVID-19 pandemic, but nevertheless, we did not notice any major problems. Sometimes, first attempts were invalid because of protocol violations. Furthermore, radiologists, especially in small hospitals, were reluctant to write a final conclusion on whether elaborated images fulfill or do not fulfill tabulated CCA criteria due to fear of making a misdiagnosis in such a specific clinical situation. Occasionally, we observed a premature CTA examination almost immediately after appearance of brainstem areflexia resulting in the presence of persisted opacification of M4 segments of the middle cerebral artery because the intracranial pressure still did not reach a value sufficient to completely block intracranial blood flow. Repeated examinations after 12 h usually confirmed CCA. Interestingly, we recognized expected and unexpected reasons for acceptance or refusal for implementation of these new technologies. In many middle-sized hospitals, in which DSA and any other brain blood perfusion studies were unavailable, CTA was relatively easily implemented as the only ancillary test facilitating and shortening of BD/DNC diagnostic process. Surprisingly, implementation process was slow in some of most advanced interventional radiology centers with permanent availability of diagnostic/interventional team. In such units, apart from strong adherence to diagnostic traditions, it was sometimes easier to organize DSA than CTA/CTP. Additionally, this was treated as a chance for training of residents in DSA procedures.

Implementation of CTP was more complicated. Not all CT scanners in Polish hospitals are able to perform whole-brain perfusion imaging. Moreover, even in well-equipped reference hospitals, usually only a few radiologists are experienced in CTP postprocessing and interpretation because in Poland, it is a relatively new technology introduced mainly in departments involved in neuroradiologic procedures. Furthermore, data postprocessing is extremely time-consuming. Due to these all difficulties, even in centers that implemented this technique in Poland, CTP examinations are usually performed within working hours and in the remainder of the week, CCA diagnosis is based on CTA.

## 7. Comparison with Recommendations Made in the World Brain Death Project

Recently Greer at al. published in JAMA great work elaborated by international group of experts, entitled “Determination of Brain Death/Death by Neurologic Criteria: The World Brain Death Project” consisting of introduction part and 17 supplements [4]. It summarizes current knowledge about various aspects of pathophysiology of brain injury leading, finally, to BD/DNC, all aspects of diagnostic procedures, possible organ procurement and, finally, future research agenda. The following is stated in it: “It is recommended that when ancillary testing is performed and demonstrates the presence of brain blood flow, BD/DNC cannot be declared at that time”. This indirectly points out the necessity for the elaboration of precise diagnostic criteria for CTA/CTP after implementation of these new technologies for investigation of CCA, both in infratentorial and supratentorial spaces. Our research results and their interpretations are consistent with this point of view.The following is stated in the “Determination of Brain Death/Death by Neurologic Criteria. The World Brain Death Project” publication: “It is recommended that when ancillary testing is performed and demonstrates the presence of brain blood flow, BD/DNC cannot be declared at that time” [4]. This indirectly points out the necessity for the elaboration of precise criteria for CTA/CTP after implementation of these new technologies for investigation of CCA, both in infratentorial and supratentorial spaces. Our research results and their interpretations are consistent with this point of view.

On the other hand, our positive opinion concerning the feasibility of CTA and CTP for CCA confirmation during BD/DNC diagnosis is discrepant with the opinion of Greer et al. [4], who stated that these methods require further consensus on the phases and timing of image acquisition, as well as consensus upon and validation of the interpretation criteria subsequently used. According to Greer, this also concerns validation in comparison to “gold-standard” BD/DNC cerebral perfusion tests such as DSA or radionuclide scintigraphy. To support our standpoint, we highlight that in our earlier publications CTA was validated against DSA [5,14,29] and in our later studies CTA/CTP were validated against clinical diagnosis supported by TCD [21,25].

In order to verify the reason of some kind of distrust of Greer at al. towards validity of CTA/CTP for CCA diagnosis we extensively analyzed Supplement 5 to “World Brain Death Project”. This supplement deals with all ancillary tests used for BD/DNC with special attention directed towards brain blood perfusion tests, including CTA/CTP. We found information about “one report of false positive result” and because of its crucial importance we meticulously analyzed the source publication [30] where we found following facts:Non contrast CT (Figure 1) in our opinion confirms severe edema in course of devastating brain injury indirectly indicating presence of severe intracranial hypertension,CTA imaging pattern is typical for CCA (Figure 2); however, the authors declare it doubtful because of possible hypotension during the procedure which is an obvious diagnostic protocol violation and makes the examination not interpretative,In TCD imaging (Figure 3) intracranial arteries might be not properly identified and flow spectra incorrectly interpreted:The typical flow spectrum in OA (ophthalmic artery) is usually different from the one showed in Figure 3. In transorbital window in TCD (transcranial Doppler) it should be higher resistive than presented on the depth 50–60 mm as OA is an artery of predominantly elastic type. Furthermore, presented flow was inconsistent with intracranial hypertension. Therefore, the flow described as the right OA perhaps does not represent true flow in OA. Regardless of these doubts the flow in OA is not a TCD criterion for CCA diagnosis. Therefore, the reason for demonstration of flow in a vessel identified as OA by default in order to support supposition of preserved cerebral perfusion remains doubtful.In patients with high intracranial pressure the flow spectra in cerebral arteries change in a very characteristic manner. The systolic phase of spectrum become very short (velocities are normal or diminished) and all diastolic velocities decline to baseline or near it. Such type of flow can persist for some time and usually leads to CCA, while in Figure 3, the flow spectrum in artery recognized by the authors as left middle cerebral artery (LMCA) is low resistant with gradual reduction of velocity during systole and diastole. This does not represent residual flow consistent with severe intracranial hypertension.

In summary, using of this publication as an argument suggesting inaccuracy of CTA in BD/DNC diagnostic procedures is questionable.

It is stated in ‘The World Brain Death Project” that “there is still no consensus on the technical criteria for CT angiography as ancillary test and considerable variation on reported sensitivity in the diagnosis of BD/DNC”. We agree with this opinion but, on the other hand, we believe that secondary analysis of available literature would help to remove some concerns. Large number of scales used and unclear information about time gap between the onset of brainstem areflexia and proceeding of CTA are highly confusing. However, if we restrict our attention to reports providing detailed information about time gap between the onset of brainstem areflexia and CTA, the data look more optimistic. Kerhuel using French criteria based on 4-point score showed that short time between clinical brain death and CTA leads to higher number of inconclusive results (low sensitivity) and postulated that time delay > 6 h provides sensitivity of 92% [16]. Similar tendency was reported by Welschehold [13]. This clearly points out that proper timing is a crucial factor determining CTA sensitivity regardless of protocol used and that minimal time delay should be recommended in international and national guidelines.

We would like to highlight that in Europe three detailed instructions are currently used, French [15] German [18] and Polish [3], based on earlier extensive research completed in these countries [6,7,8,10,11,12,13,14,21,22,25,28,29]. Interestingly, a proposition of earlier TCD was included in French instruction to minimize delay from the onset of brainstem areflexia to CTA and to avoid premature examination [15]. TCD, even uncertified or Duplex Doppler examination of both vertebral arteries and both internal carotid arteries in the extracranial segments may be performed to determine a proper time for CT tests We included this proposition in the recently initiated Polish trial concerning validation CTA for CCA diagnosis in the pediatric population [28]. In addition, we are planning to recommend Duplex Doppler as pretest before CTA in future amendment of Polish BD/DNC criteria. Summarizing our considerations concerning influence of time delay on CTA sensitivity, we presume that prospective great trial currently conducted by Chassé and Shankar in Canada will answer the majority of concerns.

Furthermore, we would like to comment on another inaccuracy we found in “World Brain Death Project”, creating negative opinion about the validity of CTA/CTP in BD/DNC procedures and simultaneously about our research results concerning our first article published in 2010 [5]. In Table 4 of Supplement 5, the “Polish scale” was cited as a source of information, with a reported sensitivity of 41.7%. This was incorrect, because in this paper, we only demonstrated an observed opacification level of all vessels examined in a group of 24 patients with confirmed BD/DNC. We neither validated the results according to any scoring scale, nor reported any sensitivity in it. Therefore, the term “Polish scale”, as well as the information about sensitivity of 41.7%, is not supported by presented data. Finally, we did not suggest the re-addition of anterior and posterior circulation assessments to the 4-point scale. However, if the data of this small group presented in Table 1 were analyzed using a 4-point scale, we would obtain a sensitivity of 100%.

The last inaccuracy on which we would like to comment concerns a source of Figure 4 visualizing “variation in methods of assessing brain blood flow on CTA, depending on choice of anatomical vasculature and time of imaging” which was of our authorship [14], but incorrectly presented as originating from the publication of the other authors. We tried to correct this erroneous information concerning our publications in a letter to editor, but unfortunately, it was not accepted because of “space limitations in the letters section”.

## 8. Conclusions

Based on our experience, results of our investigations, extensive literature review and, finally, recent observations and confidential discussions with diagnostic teams we conclude:CTA and CTP-derived CTA might be in future the tests of choice for CCA diagnosis due to increasing availability and relatively easy interpretation.Proper timing based on time elapse after the appearance of brain stem areflexia and/or Doppler pretest might significantly reduce preterm examinations and significantly increase sensitivity of CTA in CCA diagnosis procedures.Whole brain CTP might be decisive in some cases of inconclusive CTA.The monitoring of the implementation of CTA/CTP according to recently amended Polish BD/DNC diagnosis guidelines does not show any major obstacles, occasionally appearing teaching troubles and excessively large sticking to traditional diagnostic schemes are gradually eliminated.We strongly believe that in next edition of “The World Brain Death Project”, CTA and CTP will be recommended as ancillary tests of choice for CCA diagnosis during BD/DNC diagnosis procedures. We strongly believe that in next edition of “The World Brain Death Project” CTA and CTP will be recommended as ancillary tests of choice for CCA confirmation during BD/DNC diagnosis procedures.

## Figures and Tables

**Figure 1 jcm-10-04237-f001:**
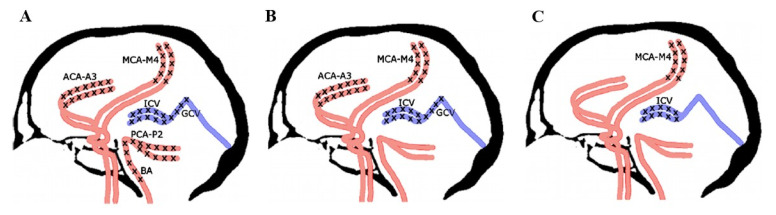
Various scales used for CCA diagnosis in CTA imaging: (**A**) 10-point scale, where positive result (score = 10) confirming CCA is stated when bilateral ACA-A3, MCA-M4, PCA-P2 and ICV and single GCV and BA are not opacified; (**B**) 7-point scale, where positive result (score = 7) confirming CCA is stated when bilateral ACA-A3, MCA-M4 and ICV and single GCV are not opacified; (**C**) 4-point scale, where positive result (score = 4) confirming CCA is stated when bilateral MCA-M4 and ICV are not opacified.

**Figure 2 jcm-10-04237-f002:**
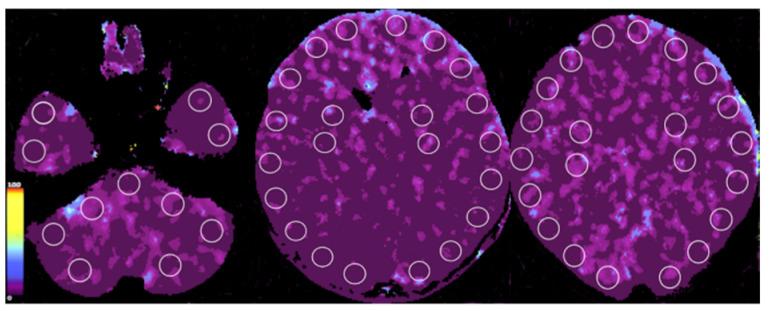
Criteria of CCA in CTP imaging. Axial sections of brain with marked positions of ROIs. Color scale illustrates range of CBF (mL/100 g/min). CBF < 10 mL/100 g/min confirms CCA.

**Figure 3 jcm-10-04237-f003:**
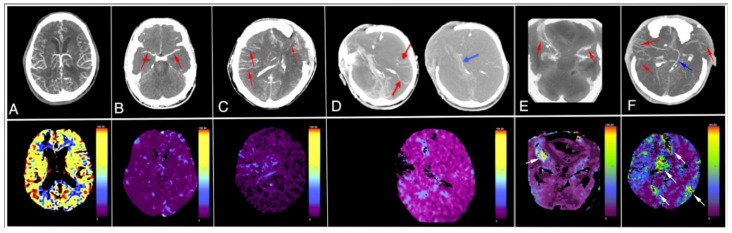
Different CTA (upper row) and CTP (lower row) patterns during CCA diagnosis procedures: (**A**) patient with suspected ischemic stroke with normal CTA and CTP; (**B**) patient with opacification limited to proximal segments of middle cerebral arteries MCA-M1 inCTA (red arrows) and CBF value < 10 mL/100 g/min in CTP; both techniques confirm CCA. (**C**) Patient with bilateral opacification of cortical arterial segments (MCA-M4; red arrows) in CTA, not consistent with CCA diagnosis and CBF value < 10 mL/100 g/min in CTP, which confirms CCA diagnosis; (**D**) patient with opacified MCA-M4 segments (red arrows) and opacified internal cerebral vein (blue arrow) in CTA, not consistent with CCA diagnosis and CBF value < 10 mL/100 g/min, which confirms CCA diagnosis; (**E**) patient with opacified MCA-M2/M3 segments (red arrows) in CTA, consistent with CCA diagnosis and isolated single sub-craniectomy area with CBF value > 10 mL/100 g/min (white arrow), also consistent with CCA diagnosis; (**F**) patient with opacified MCA-M4 segments (red arrows) and opacified internal cerebral vein (blue arrow) in CTA, not consistent with CCA diagnosis and multiple scattered areas with CBF value > 10 mL/100 g/min (white arrows), also inconsistent with CCA diagnosis. Color scales illustrate range of CBF (mL/100 g/min). CBF < 10 mL/100 g/min confirms CCA.

**Figure 4 jcm-10-04237-f004:**
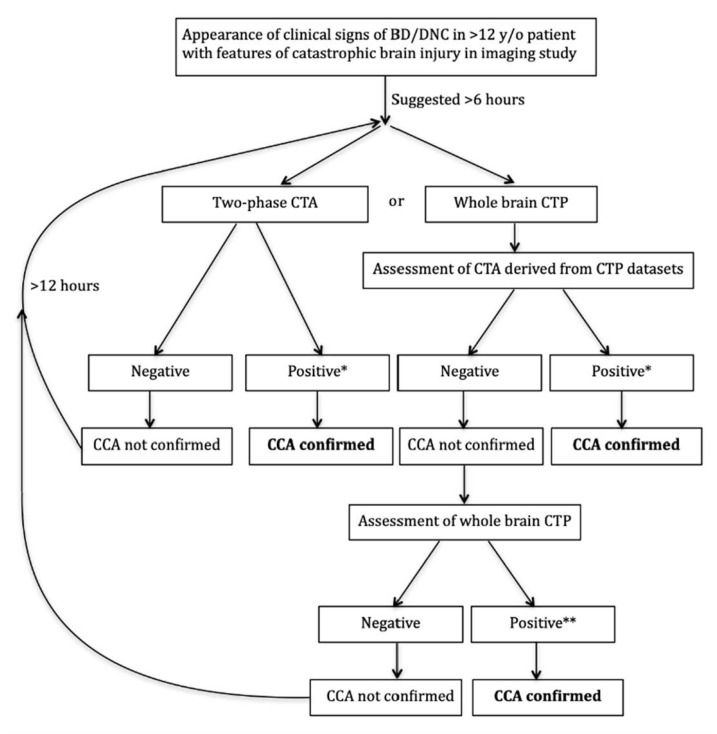
Schematic algorithm of application of CTA and CTP for CCA confirmation according to Polish national guidelines for BD/DNC diagnosis. Notes: * bilateral non-filling of cortical arteries (MCA-M4) and ICVs in late phase with normal filling of extracranial arteries in early phase; filling of one or two cortical arteries on the same side is permissible as long as the ICVs are not filled. ** CBF below 10 mL/100 g/min and CBV below 1.0 mL/100 g in all ROIs. Presence of small, isolated foci with CBF or CBV above these values are permissible in regions of local decompression due to craniectomy or open skull fracture. BD/DNC—brain death/death by neurologic criteria; CCA—cerebral circulatory arrest.

**Table 1 jcm-10-04237-t001:** Comparison of three frequently used European national guidelines for determination CCA with CTA during BD/DNC diagnosis procedures.

	Polish (2020 *)	French (2011 *)	German (2015 *)
Recommended delay after appearance of clinical signs of BD/DNC (h)	6	6 **	not specified
1. Non-contrast scanning used as a reference			
2. Early post-contrast scanning			
Contrast volume (mL)	80	2 mL/kg (max 120)	65
Scanning time	triggered by bolus-tracking in extracranial carotid arteries	20 s after start of contrast injection	not performed
Assessed vessel			
STA (bilaterally) ***	2	2	not performed
3. Late post-contrast scanning			
Scanning time	40 s after start of early post-contrast scanning	60 s after start of contrast injection	15 s after filling of extracranial carotid arteries detected with bolus-tracking
Evaluation scale	4-point	4-point	7-point late arterial
Assessed vessel			
STA (bilaterally) ***			2
MCA-M1 (bilaterally)			2
ACA-A1 (bilaterally)			2
BA			1
PCA-P1 (bilaterally)			2
MCA-M4 (bilaterally)	2	2	
ICV (bilaterally)	2	2	
Delay to next exam if the previous was inconclusive (h)	12	not specified	not specified

Notes: BD—brain death; STA—superficial temporal artery; ACA—anterior cerebral artery; A1—1st division of ACA; MCA—middle cerebral artery; M1—1st division of MCA; M4—4th division of MCA; PCA—posterior cerebral artery; P1—1st division of PCA; BA—basilar artery; ICV—internal cerebral vein. * year of implementation. ** time delay of 6 h can be shortened by performing transcranial Doppler ultrasound. *** assessment of filling of extracranial arteries like STA serves as a control of effective contrast administration to the head.

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
