# Peer review of "Implementation of Computed Tomography Angiography (CTA) and Computed Tomography Perfusion (CTP) in Polish Guidelines for Determination of Cerebral Circulatory Arrest (CCA) during Brain Death/Death by Neurological Criteria (BD/DNC) Diagnosis Procedure"

_jcm, 2021, doi:10.3390/jcm10184237_

Round 1

Reviewer 1 Report

Ensuring a reliable diagnosis of death by neurological criteria is an important target, particularly when an ancillary confirmatory test is advisable or mandatory. Ideal procedures, using available cerebral blood flow methods, have not been identified yet; consequently, scientific evidence about sensitivity and potential role of computed tomography perfusion (CTP) is strongly desirable.

In this paper the Authors describe how CTP has been evaluated in clinically reliable brain deaths(BD/DNC), and validated as confirmatory test in potentially confounding situations, particularly related to opacification of intracranial arteries  (shown by digital subtraction angiography/computed tomography angiography) due to “stasis filling”, leading to refused/missing BD diagnosis.

The aim of the paper was “to present the evolution of Polish guidelines for BD/DNC diagnosis and to demonstrate scientific evidence supporting the implementation of computed tomography angiography (CTA) and computed tomography perfusion (CTP) for confirmation of cerebral circulatory arrest (CCA).”

Unfortunately, novel scientific studies and results are not included in the paper; only previously published data  by the same Polish group (Sawicki et al. Clin Neuroradiol 29:101-108, 2019 and other papers cited as references) or “unpublished observations” are reported. Figures are not original.

The structure of the paper is not appropriate to a scientific article (background, methods, results, discussion,  conclusions). Recent international recommendations and suggestions regarding ancillary tests for BD diagnosis  (Greer et al. JAMA 2020: It is suggested that some priority be given to the further validation of CTA, given its increasing prevalence and usage. Integration with CT perfusion may prove valuable, given recent advances in CT technology.”) should be discussed and, in my opinion, fulfilled. These recommendations underline the good reasons for improving the scientific background of CT Perfusion in this context.

Nevertheless, the topic of the paper is relevant and the message for international clinical practitioners is important, particularly as a review of the scientific, clinical, organizational and “political” pathway towards implementing a new test (CTP as a useful and tailored adjunct in the diagnosis on brain death)  in  national  guidelines. I would suggest focusing on the appropriate use of CTA plus CTP  and the history of the Polish successful process. Desirable consequences should be international projects aimed to wide validation of standardized CTP methodology and criteria in adults and pediatrics.

Author Response

  1. Comment: Unfortunately, novel scientific studies and results are not included in the paper; only previously published data by the same Polish group (Sawicki et al. Clin Neuroradiol 29:101-108, 2019 and other papers cited as references) or “unpublished observations” are reported. Reply: As we intended to demonstrate the pathway for implementation of detailed Polish instruction for CTA/CTP usage for BD/DNC we had to follow step by step development of our experience and our parallel publications.. We felt that cited older and new publications of the other authors created sufficient background for it. Unpublished observations and personal communications are sometimes reported in scientific articles especially if they were collected in 40 years and assistance in 1500-2000 procedures. Nevertheless, we widened the reference list including  papers used for development of French and German guidelines.
  2. Comment: Unfortunately, novel scientific studies and results are not included in the paper. Reply: According to suggestion we supplemented novel scientific studies.
  3. Comment: Figures are not original. Reply: This comment is unclear to us. All CT images originate from our database. Pictures were drawn by us and originally presented in slightly different form in Sawicki, M.; Bohatyrewicz, R. et al. Computed tomographic angiography criteria in the diagnosis of brain death—Comparison of sensitivity and interobserver reliability of different evaluation scales. 2014, 56, 609–620, doi:10.1007/s00234-014-1364-9. Their origin was incorrectly cited in „World Brain Death Project”.
  4. Comment: The structure of the paper is not appropriate to a scientific article (background, methods, results, discussion, conclusions). Reply: we changed the qualification of article in to narrative review which better corresponds with it’s contents.

Reviewer 2 Report

General Comments

  • In this paper Bohatyrewicz and colleagues cement Poland’s place as the world leader for CTA/CTP in diagnosing BD/DNC. This is especially important given the negativity toward this technique in the World Brain Death Project.
  • I found the way the paper was structured odd, with a one paragraph introduction, one paragraph aims and then into results with no methods. It isn't results you are sharing but the history of your progress in using CTA/CTP. I offer a suggestion below of, what I think, would be a stronger and clearer structure.
  • Suggest you bring some of the discussion of the world brain death project (Greer et al) into the early part to set the scene and then create a separate section at the end to answer their criticism with your evidence. Greer et als paper is pivotal for framing your paper. It needs to be used more clearly and deliberately.
  • Apart from the odd structure, very readable. The science is strong. Diagrams helpful. Thank you.
  • You talk a lot about sensitivity but it is specificity the clinical team needs more. If we say no flow on CTA/CTP how confident can we be. This is more important than if there is no flow how sensitive are we in detecting it. The safety of the diagnosis means we want to be 100% specific. The difference between sensitivity and specify needs to be brought out. I acknowledge that nearly all papers on this topic all talk about sensitivity as non-suspected brain dead patients are usually not involved in the research. But this needs to be explained to the reader.
  • Can you please more clearly explain the Polish criteria for CTA/CTP – do you have to do CTP as well or is CTA alone ok? Please help us understand by summarising into a paragraph (or a table) what your criteria actually are.

Minor Comments

“Abstract: Introduction: Brain death/death by neurologic criteria (BD/DNC) guidelines are routinely analyzed, compared and updated in the majority of countries and are later novelized as national criteria.” Implemented is better than novelized – which implies fiction.

“The first Polish BD/DNC criteria implemented in 1884” I think you meant 1984?

“Despite receiving the official support of health care authorities and three consecutive Popes, we have faced unclear or even negative statements from some ultraconservative religious circles and ultraconservative Catholic media.”

- stay more neutral “Despite receiving the official support of health care authorities and three consecutive Popes, we have faced unclear or even negative statements from some religious commentors and selected Catholic media.”

Proposed alternate structure for the paper

Please note I have not corrected for other comments above.

This is how I would suggest the paper is structured in its current format, with suggested subheadings to aid clarity and readability.

In this narrative review, we present the evolution of Polish guidelines for BD/DNC diagnosis and to demonstrate scientific evidence supporting the implementation of computed tomography angiography (CTA) and computed tomography perfusion (CTP) for confirmation of cerebral circulatory arrest (CCA).

Background

The first Polish BD/DNC criteria implemented in 1884 as whole-brain death criteria were similar to the so-called “Harvard brain death criteria” [1]. Later, these Polish criteria underwent a unique evolution: in 1990, they were converted into brainstem death criteria, and later in 2007, they were reversed back to whole-brain death criteria [6], subsequently being amended in 2020 [7]. During this conversion, we reimplemented an opportunity for facultative usage of instrumental ancillary tests, including brain blood perfusion tests.

Currently in Poland, although formally facultative, instrumental ancillary tests are used in the majority of BD/DNC diagnosis procedures. This is exactly the reverse of what is proposed in current international recommendations [5]. There are various reasons for this. Despite receiving the official support of health care authorities and three consecutive Popes, we have faced unclear or even negative statements from some ultraconservative religious circles and ultraconservative Catholic media. The matter is sometimes compli- cated by political activities, which induce doctors’ uncertainty and filling of a lack of legal safety. Polish physicians would feel more comfortable and safe if they could demonstrate the results of instrumental ancillary tests confirming BD diagnosis. Brain blood perfusion tests are considered to be the most evident from all sides of the discussion—physicians, relatives of the deceased and public opinion. It is obvious to everybody, regardless of the level of education, that non-perfused tissue must be dead.

There are three brain blood perfusion ancillary tests that, despite known disad- vantages, have a long-established position in BD/DNC diagnosis:

  1. Catheter digital subtraction angiography (DSA), selective or from the aortic arch;
  2. Transcranial Doppler ultrasonography (TCD);
  3. Cerebral perfusion scintigraphy.

Use of DSA, which is still considered to be a gold standard and reference method, is gradually decreasing [5]. It is invasive and requires procedural skills, time and availability of an angiography suite. It was reimplemented in Polish BD criteria in 2007 and was ap- proved again in the amendment in 2020 [7]. TCD is completely noninvasive, its sensitivity exceeds 90% and the specificity is 100%. Despite its obvious advantages, TCD is not fre- quently used in Poland because it has to be performed with unique devices that, currently, are not widely available. Additionally, it is highly operator-dependent and requires certi- fication of the performing physician. The last of the “traditional” cerebral perfusion tests, scintigraphy, is currently seldom available in the majority of Polish hospitals.

In this difficult situation, it has become of crucial importance to develop and intro- duce an ancillary test that would be less invasive than DSA, easily available, uncompli- cated to perform and easy to interpret.

Early research into using CTA/CTP for diagnosing BD/DNC in Poland

In 2005, we considered CTA and CTP to be the tests of choice for the future, which is consistent with the current opinion of Greer et al. [5]. Modern multi-slice CT scanners are fast enough to visualize vasculature and perfusion of the whole brain with a single intravenous injection of iodinated contrast medium and, finally, to confirm CCA. Both CTA and CTP, if performed with a calibrated intravenous contrast injection and precise scanning protocols, are operator-independent at the stage of performance and providing raw data.

The research team of the Department of Anesthesiology and Intensive Care together with co-workers from the Department of Diagnostic Imaging and Interventional Radiol- ogy of the Pomeranian Medical University in Szczecin have been involved in research programs and legislation since the implementation of the first Polish BD/DNC criteria published in 1984. In 2005–2007, Romuald Bohatyrewicz was a co-chairman, and in 2015– 2019, chairman, of the Ministry of Health’s Task Force for review of these criteria.

In 2007, we suggested the possibility of implementing CTA as a new brain blood perfusion test, but this was not accepted by the rest of the Task Force members because of insufficient evidence in the literature and lack of experience in Poland. In this situation, we organized a national multi-center trial (N N403 171137), entitled “Evaluation of CT angiography and CT perfusion in brain death diagnosis”, in a group of adult brain-dead patients, followed by a series of publications. The first of them, published in 2010 [8], con- firmed the ability of CTA/CTP to diagnose CCA in our population of BD/DNC patients, and our findings were compatible with data published by other authors at that time [9– 11]. Unfortunately, our CTP findings could not be applied to CCA diagnosis because the generation of CT scanners used at that time in Poland could cover only a thin layer of the brain, about 30 mm thick.

During initial attempts to implement CTA for CCA diagnosis, there was no consen- sus regarding the evaluation criteria, which have evolved as research progressed [10–13]. The most popular scoring systems (10, 7, 4 points), shown in Figure 1, were based on anal- ysis of the opacification of the following:

  1. Pericallosal segments of the right and left anterior cerebral artery (ACA-A3);
  2. Cortical segments of the right and left middle cerebral artery (MCA-M4);
  3. Cortical segments of the right and left posterior cerebral artery (PCA-P2);
  4. Basilar artery (BA);
  5. Right and left internal cerebral vein (ICV);
  6. Great cerebral vein (GCV)—the vein of Galen.

Initially, we analyzed CTA imaging in a group of 82 patients undergoing routine BD/DNC diagnosis with DSA included as a standard element of this procedure. CTA was completed first and was followed by DSA [14]. In this situation, CTA could be validated against DSA with a very short time interval between these two procedures, which is con- sistent with the recommendation for method validation recently published by Greer et al. [5]. The sensitivity reported in this study reached 96.3% according to the 4-point scale, 74.4% according to the 7-point scale and 67.1% according to the 10-point scale [14].

After meticulous analysis of these data, we finally accepted the 4-point scale pro- posed by the French guidelines for diagnosis of BD/DNC [15]. According to this 4-point scale, CCA may be confirmed if there is a bilateral absence of contrast filling of cortical segments of the middle cerebral arteries (MCA-M4) and internal cerebral veins (ICVs), as presented in Figure 1. Unilateral opacification of one or two cortical branches of the MCA does not preclude the diagnosis of CCA as long as the contrast does not fill the ICVs.

Research advancement with CTP in Poland

At the same time, we noticed that in rare cases of patients demonstrating BD/DNC symptoms, preserved trace opacification of intracranial arteries may be observed in DSA examination. This phenomenon is known as stasis filling, defined as delayed, weak and persistent opacification of the proximal cerebral arterial segments, without opacification of the cortical branches or venous outflow [16]. In these rare cases, CTP often shows re- sidual cerebral blood flow (CBF) below 10 mL/100 g/min and a cerebral blood volume (CBV) below 1.0 mL/100 g. These values are the established thresholds for neuronal ne- crosis [17], and following this, they may be considered thresholds for global or regional CCA diagnosis [11,18,19].

Advanced research activity concerning CTP became possible after the advent of a new generation of CT scanners fast enough to visualize vasculature and perfusion of the whole brain with a single intravenous injection of iodinated contrast medium. At that time, we stopped performing CTA imaging and switched to reconstruction of CTA images from the CTP source images as timing-invariant (TI)-CTA. TI-CTA provides angiography by overlapping all time frames and displaying the maximum enhancement over time. This makes the technique time independent, which means that the maximum enhancement of a vessel is displayed independently of contrast arrival time. Therefore, TI-CTA is not sen- sitive to delayed arrival of the contrast material in cerebral vessels and, thus, should dis- play any vessel present. This technique was previously described and shown to be reliable by Smit et al. [20].

CTP criteria for CCA during the BD/DNC diagnostic procedure were not published before; therefore, we elaborated our original instruction of assessment based on an analy- sis of the CBF and CBV in 1-cm2 circular regions of interest (ROIs), including the midbrain (two ROIs), the pons (two ROIs) and the medulla oblongata (two ROIs) as well as the cerebellum (eight ROIs); cortical regions of the frontal (12 ROIs), parietal (12 ROIs), tem- poral (12 ROIs) and occipital lobes (eight ROIs); and the basal ganglia (eight ROIs), drawn bilaterally and placed on each 10-millimeter axial slice, as shown in Figure 2. We recog- nized CCA in CTP examination if the CBF value was below 10 mL/100 g/min and CBV was below 1.0 mL/100 g in all ROIs [19]. The most frequent combinations of CTA and CTP images are shown in Figure 3.

In the next step of our research program, we hypothesized that CTP would be a more sensitive approach than CTA in CCA diagnosis. To verify this, we conducted a study aim- ing to compare the sensitivities of CTP and CTA in recognizing CCA during BD/DNC diagnosis procedures. A group of 50 patients undergoing this diagnostic procedure were included in the study. All of them met the standard BD/DNC criteria based on confirma- tion of catastrophic brain injury, exclusion of confounders and confirmation of brainstem areflexia and apnea during two series of clinical examinations (Sawicki 19). Additionally, TCD examination confirming CCA was completed in the majority of them; however, this information was not included in the publication.

In 43 out of 50 patients, CTA confirmed CCA, as demonstrated in Figure 3, patient B. In the remaining seven patients, CTA revealed opacification of M4 segments, ICVs or both, as shown in Figure 3, patients C and D. These CTA findings were inconsistent with CCA according to the 4-point scale. In all 50 patients, CBF was below 10 mL/100 g/min and CBV was below 1.0 mL/100 g, which confirmed CCA according to our criteria. In summary, in this publication, we reported a sensitivity of 86% and 100% for CTA and CTP, respectively. Additionally, these results confirmed our hypothesis that in borderline cases, when CTA is inconclusive, CTP may be a decisive method for CCA diagnosis. Later, after analysis of a few additional cases, we stated that in special situations in patients with clinical signs of BD, isolated areas of decompression may be preserved in the region of craniectomy or open fractures. These isolated areas may exhibit CBF and CBV values above the thresh- olds of 10 mL/100 g/min and 1 mL/100 g, respectively. This phenomenon does not exclude the diagnosis of CCA if the CBF and CBV values are below these thresholds in other ROIs, including those in the brainstem, as shown in Figure 3, patient E. Appearance of multiple areas with CBF and CBV above the threshold values is, according to our current opinion, inconsistent with CCA diagnosis, as shown in Figure 3, patient F.

Additionally, we noticed (unpublished results) that if CTA tests were performed shortly after the appearance of brainstem areflexia, more widespread opacification of cer- ebral vessels, thus excluding CCA diagnosis, was recorded, which was also confirmed by the data published by Kerhuel et al. [21]. The explanation for this phenomenon is quite simple: in this short period, the intracranial pressure (ICP) did not exceed the mean arte- rial pressure (MAP), leading to CCA. Premature CTA examinations lead to a dramatic decrease in test sensitivity, which is one of the sources of undeserved opinions about the poor utility of this method for CCA determination. Such cases are rather unlikely in Po- land because our doctors prefer more conservative approaches and generally initiate di- agnostic procedures after a longer observation period. Nevertheless, in the Polish guide- lines [7], we recommended a minimal 6-hour observation time before CTA/CTP examina- tions, which is identical to the French guidelines [15].

Noteworthily, another research group led by Shankar [18,22] postulated a slightly different approach to the possible usefulness of CTP imaging in BD/DNC diagnostic pro- cedures or withdrawal of life-sustaining therapy. They focused on demonstration of brain hypoperfusion limited to the brainstem area, which is, in fact, somehow parallel to brain- stem areflexia and the brainstem death concept, suggesting “isolated brainstem death”. CTP in this situation does not necessarily confirm global CCA as it was demonstrated in our report [19]. It is highly questionable whether isolated brainstem death diagnosis con- firmed by CTP imaging, but coexisting with preserved supratentorial perfusion and pos- sibly persisted EEG activity, might justify BD/DNC diagnosis.

Implementation of CTA/CTP examination into the Polish national guidelines for BD/DNC

Based on our published research data [14,19] and unpublished observations, we fi- nally implemented CTA/CTP examination in the Polish national guidelines for BD/DNC diagnosis in patients over 12 years of age at the beginning of 2020 [7]. To our knowledge, this was the first implementation of CTP in official BD/DNC diagnosis guidelines.

CTA/CTP examinations for BD/DNC diagnosis procedures were approved for pa- tients >12 years old, assuming that above this age, the skull is not pliable and the brain reaches morphological maturity. We presumed that the mechanisms of CCA might be similar in younger age groups, at least >2 years old without patent sutures or fontanels, and in patients <2 years old, according to our knowledge, it is unpredictable, as with the results of other traditional methods used for CCA diagnosis [5]. Therefore, in order to explore this issue, we recently invited all Polish and foreign pediatric intensive care units to participate in a multi-center study for the validation of CTA and CTP in determination of CCA during the BD/DNC diagnosis procedure in a pediatric population below 12 years of age [23].

Immediately after the introduction of CTA/CTP for CCA diagnosis into Polish guide- lines, we started monitoring the usage of these methods all over the country. Unfortu- nately, this was extremely difficult due to the COVID-19 pandemic, but nevertheless, we did not notice any major problems. Sometimes, first attempts were invalid because of pro- tocol violations. Furthermore, radiologists, especially in small hospitals, were reluctant to write a final conclusion on whether elaborated images fulfill or do not fulfill tabularized CCA criteria due to fear of making a misdiagnosis in such a specific clinical situation. Occasionally, we observed a premature CTA examination almost immediately after ap- pearance of brainstem areflexia resulting in the presence of persisted opacification of M4 segments of the middle cerebral artery because the intracranial pressure still did not reach a value sufficient to completely block intracranial blood flow. Repeated examinations after 12 h usually confirmed CCA.

Implementation of CTP was more complicated. Not all CT scanners in Polish hospi- tals are able to perform whole-brain perfusion imaging. Moreover, even in well-equipped reference hospitals, usually only a few radiologists are experienced in CTP postprocessing and interpretation because in Poland, it is a relatively new technology introduced mainly in departments involved in neuroradiologic procedures. Furthermore, data postpro- cessing is extremely time-consuming. Due to these all difficulties, even in centers that im- plemented this technique in Poland, CTP examinations are usually performed within working hours, and in the remainder of the week, CCA diagnosis is based on CTA.

Comparison with Recommendations made in the World Brain Death Project

The following is stated in the “Determination of Brain Death/Death by Neurologic Criteria. The World Brain Death Project” publication: “It is recommended that when ancillary testing is performed and demonstrates the presence of brain blood flow, BD/DNC cannot be declared at that time” [5]. This indirectly points out the necessity for the elaboration of precise criteria for CTA/CTP after implementation of these new technologies for investigation of CCA, both in infratentorial and supratentorial spaces. Our research results and their interpretations are consistent with this point of view.

On the other hand, our positive opinion concerning the feasibility of CTA and CTP for CCA confirmation during BD/DNC diagnosis is discrepant with the opinion of Greer et al. [5], who stated that these methods require further consensus on the phases and tim- ing of image acquisition, as well as consensus upon and validation of the interpretation criteria subsequently used. According to Greer, this also concerns validation in compari- son to “gold-standard” BD/DNC cerebral perfusion tests such as DSA or radionuclide scintigraphy. To support our standpoint, we highlight that in our first basic publication, CTA was validated against DSA [14], and in the second one, CTA/CTP were validated against clinical diagnosis supported by TCD.

Expand and compare more – this is important – why are you right and them worng.

Conclusions

The monitoring of the implementation of CTA/CTP in the recently amended Polish BD/DNC diagnosis guidelines does not show any major obstacles. CTA/CTP seem to be superior to the methods traditionally used for CCA diagnosis, and their implementation in routine diagnostic procedures is currently taking place, with teaching troubles being gradually eliminated.

This conclusion could be stronger

Author Response

  1. Comment: I found the way the paper was structured odd, with a one paragraph introduction, one paragraph aims and then into results with no methods. It isn't results you are sharing but the history of your progress in using CTA/CTP. I offer a suggestion below of, what I think, would be a stronger and clearer structure. Reply: the structure of manuscript was modified according to suggestion and qualification of manuscript was changed in to narrative review
  2. Comment: Suggest you bring some of the discussion of the world brain death project (Greer et al.) into the early part to set the scene and then create a separate section at the end to answer their criticism with your evidence. Greer et al. paper is pivotal for framing your paper. It needs to be used more clearly and deliberately. Reply: according to suggestion we created special section for polemic with some aspects of “Word Brain Death Project” and pointed out some inaccuracies depreciating the value CTA/CTP tests.
  3. Comment: You talk a lot about sensitivity but it is specificity the clinical team needs more. If we say no flow on CTA/CTP how confident can we be. This is more important than if there is no flow how sensitive are we in detecting it. The safety of the diagnosis means we want to be 100% specific. The difference between sensitivity and specify needs to be brought out. I acknowledge that nearly all papers on this topic all talk about sensitivity as non-suspected brain dead patients are usually not involved in the research. But this needs to be explained to the reader. Reply: We supplemented special  subsection concerning specificity in  BD/DNC diagnosis.
  4. Comment: Can you please more clearly explain the Polish criteria for CTA/CTP – do you have to do CTP as well or is CTA alone ok? Please help us understand by summarising into a paragraph (or a table) what your criteria actually are. Reply: We supplemented a flowchart summarizing possible variation of CTA processing dependent upon available CT facilities, skill of radiologic team and local tradition. Additionally we included table with comparison of Polish, French and German guidelines which appear to be most frequently used in Europe.
  1. Comment: Minor comments. Reply: We applied all suggestions.

This manuscript is a resubmission of an earlier submission. The following is a list of the peer review reports and author responses from that submission.